# Operation LiLi: Using Crowd-Sourced Data and Automatic Alignment to Investigate the Phonetics and Phonology of Less-Resourced Languages

**Mathilde Hutin** [1,*,†] and **Marc Allassonnière-Tang** [2,*,†]

1   LISN-CNRS, UMR 9015, Université Paris-Saclay, 91405 Orsay, France
2   CNRS, MNHN, EA, UMR 7206, Université Paris Cité, 75016 Paris, France
*   Correspondence: mathilde.hutin@lisn.upsaclay.fr (M.H.); marc.allassonniere-tang@mnhn.fr (M.A.-T.)
†   These authors contributed equally to this work.

**Abstract:** Less-resourced languages are usually left out of phonetic studies based on large corpora. We contribute to the recent efforts to fill this gap by assessing how to use open-access, crowd-sourced audio data from Lingua Libre for phonetic research. Lingua Libre is a participative linguistic library developed by Wikimedia France in 2015. It contains more than 670k recordings in approximately 150 languages across nearly 740 speakers. As a proof of concept, we consider the Inventory Size Hypothesis, which predicts that, in a given system, variation in the realization of each vowel will be inversely related to the number of vowel categories. We investigate data from 10 languages with various numbers of vowel categories, i.e., German, Afrikaans, French, Catalan, Italian, Romanian, Polish, Russian, Spanish, and Basque. Audio files are extracted from Lingua Libre to be aligned and segmented using the Munich Automatic Segmentation System. Information on the formants of the vowel segments is then extracted to measure how vowels expand in the acoustic space and whether this is correlated with the number of vowel categories in the language. The results provide valuable insight into the question of vowel dispersion and demonstrate the wealth of information that crowd-sourced data has to offer.

**Keywords:** vowel dispersion theory; Lingua Libre; phonetics; crowd-sourced data

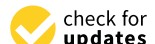



## 1. Introduction

Phonetic studies have experienced unprecedented progress with the emergence of large, automatically processable corpora. However, this progress has deepened inequalities among languages, as advances concern mostly a handful of languages, that can be defined as "well-resourced", as opposed to less-resourced languages, for which very large corpora are rare or non-existent or for which the technologies are not as efficient. Less-resourced languages are thus usually left out of large-corpus studies, although some efforts are being made, e.g., with the CMU Wilderness Corpus (Black 2019) and its offspring, the VoxClamantis corpus (Salesky et al. 2020), or the Common Voice project by Mozilla (https://commonvoice.mozilla.org, accessed on 30 August 2022) (Ardila et al. 2020) and its extension, the VoxCommunis corpus (Ahn and Chodroff 2022). The problem is not only having access to data, but also processing it, as automatic speech recognition (ASR) systems need resources to function, even though, again, some efforts have been made to train ASR systems on limited corpora (e.g., the Zero Resources Speech Challenge Dunbar et al. 2017, 2019, 2020; Versteegh et al. 2015).

Questions in phonetic typology in particular suffer from this imbalance: Data-driven research, in particular in phonetics, requires massive amounts of oral data, which can be difficult to access; Such research in a typological survey needs such data in not one, but in a significant number of languages to draw meaningful conclusions. Yet typological studies

are crucial to test questions in theoretical linguistics and to better understand language—in itself and as a human cognitive ability.

A well-known typological question lies in the vowel dispersion theory (VDT), which relies on the H&H ("Hypo- and Hyperspeech") model of communication (Lindblom 1990) and proposes that the structure of vowel inventories in the world's languages derives from the constant negotiation by speakers of functional principles, i.e., the minimization of effort on the production side and the maximization of contrast on the perception side (Liljencrants and Lindblom 1972; Lindblom 1986; Schwartz et al. 1997; Vaux and Samuels 2015; Zhang and Gong 2022). One prediction this theory makes is that, in synchrony, in a given system, variation in the realization of each vowel will be inversely related to the number of vowel categories. For instance, each vowel in a language with a dozen lexical vowels would display less internal variation (i.e., would expand within a smaller acoustic space) while those in a language with only three lexical vowels would display more internal variation (i.e., would expand within a larger acoustic space), for example with [a] realizations sounding like [e] or [o]. In the following, we refer to this hypothesis as the Inventory Size Hypothesis (ISH).

Typically, such a hypothesis can only be tested if the vowel inventory of a given language has been studied and reported, which limits the scope of the analyses. Several studies therefore propose to tackle this question by investigating a limited number of languages, with contradictory outcomes. On the one hand, a study on American English, Greek and German (Jongman et al. 1989), another study on French and two dialects of Arabic Al-Tamimi and Ferragne (2005), a third one on three German languages (Peters et al. 2017) and a fourth one on Quebec French and Inuktitut (Larouche and Steffann 2018) provide support in favor of the ISH. On the other hand, studies on English and Spanish (Bradlow 1995), on English, Spanish and French (Meunier et al. 2003), on five dialects of Catalan (Recasens and Espinosa 2009), on five dialects of Chinese (Lee 2012) and on three German languages (Heeringa et al. 2015) do not provide evidence in favor of the ISH, which can however be due, for the last three at least, to the genetic and typological closeness of the languages under survey and probable diglossy of the speakers tested. However, if the information of vowel space could be automatically extracted from audio corpora with an acceptable level of accuracy, the scope of the analyses could be significantly extended. The direct use of corpora would allow the extraction of frequency information, which can be used to test the hypothesis in depth. Such studies have been proposed in the past, i.e., on professional corpora: Engstrand and Krull (1991) found inconclusive results on 7 languages across 6 language families, while Livijn (2000) on 28 languages, Gendrot and Adda-Decker (2007) on 8 languages across 4 families, and Salesky et al. (2020) on 38 languages across 11 families, found no evidence for an effect of inventory size on the global acoustic space.

The present study has two main goals. First and foremost, we propose to contribute to the recent efforts towards less-resourced languages by exploring how it is now possible to use open-access, crowd-sourced audio data to investigate their phonetics and phonology. In this particular preliminary study, we use the data from Lingua Libre, an open access, crowd-sourced linguistic library from Wikimedia France (https://lingualibre.org/, accessed on 30 August 2022), and the open access alignment tool WebMAUS, the Munich Automatic Segmentation System (Kisler et al. 2017; Schiel 1999), to investigate a sample of 10 languages, 8 of which are less-resourced. Our ambition is that the initiation of such practices would allow the scientific community to study larger numbers of languages, most of which are usually understudied because they are under-represented in oral corpora, thus opening promising avenues for the investigation of typological comparisons and broader theoretical questions. The second goal of the paper is to apply this new methodology to the question of vocalic variation with regards to inventory size (ISH), and thus demonstrate the extent to which our methodology can be useful to provide a new insight on old questions. In particular, we will focus on the three cardinal vowels /i, a, u/, that are common to a majority of languages (Schwartz et al. 1997), and show that several types of analyses of

10 languages ranging from 5 to 14 vowels provide no evidence for the validity of the ISH, thus aligning with the most recent works relying on 7 languages or more.

The paper is organized as follows. In Section 2, we present the corpus, segmentation tool and resulting data sets for the 10 languages we selected, as well as justify some methodological choices. In Section 3, we present the results of several types of data visualization methods and of statistical analyses. In Sections 4 and 5, we discuss the results and conclude.

## 2. Materials and Methods

To test our methodology, we use the data from Lingua Libre (https://lingualibre.org, accessed on 30 August 2022), a participatory linguistic media library developed by Wikimedia France and launched in 2015, with the aim to counter the lack of oral data as well as of diversity in the languages represented on the Internet. As a crowd-sourcing tool, any speaker can log into Lingua Libre, fill in a profile with basic metadata, and record themselves (or guests) reading lists of words in their native language. The device detects pauses, which allows for the recording to end when the word has been read and the next recording to start automatically after, therefore effortlessly generating relatively short audio files for each word. Each audio file contains the recording of a word and is supposed to be titled on the same template of 'Language-Speaker name-Item name'. For example, for the recording 'fra.-Guilhelma-avril.wav', the language of the recording is French ('fra'), the speaker ID is 'Guilhelma', and the recorded item is 'avril', which means April. In February 2022, the Lingua Libre library contained almost 677,000 recordings in 147 languages across 737 speakers. Among them, 34 languages displayed at least 1000 words, across a varied number of speakers. As an open-access, crowd-sourced tool, it has the advantage of offering an ever-growing data set. To the best of our knowledge, however, besides our own research project, Lingua Libre was used only once for academic purposes, i.e., to estimate the transparency of orthographies in 17 languages using an artificial neural network (Marjou 2021).

The workflow for data extraction is as follows. First, the recordings are selected from the Lingua Libre database[1]. In the present study, we extract a subsample of 50 items for each vowel in each language, to counter the fact that some languages have many more data points than others. The threshold 50 was chosen based on the minimum count of the targeted vowels in a sample of 1000 recordings for each of the selected languages. We also aim at having a similar number of speakers approximately 10 for each language. Second, the recordings are segmented and aligned using WebMAUS (Kisler et al. 2017). WebMAUS is the online open-access version of the Munich AUtomatic Segmentation (MAUS) software (Schiel 1999), which is used to automatically time-align a recording based on its orthographic transcription. In general, MAUS creates a pronunciation hypothesis graph based on the orthographic transcript of the recording using a grapheme-to-phoneme converter. During this process, the orthographic transcription is converted to the Speech Assessment Methods Phonetic Alphabet (SAMPA (Wells 1997)). The signal is then aligned with the hypothesis graph and the alignment with the highest probability is chosen. As an overview of its accuracy, experiments have shown that the MAUS-based alignment matches human-based alignments 95% of the time (Kipp et al. 1997). Third, the selected vowels are extracted from the recordings. Finally, the extracted recordings of the selected vowels are analyzed in terms of formants. For each recording of each vowel, the F1 and F2 at the middle of the entire sound are extracted. The formants at the middle of the sound (measured on a window length of 25 milliseconds) are considered to attenuate the influence of context-induced noise in the recordings. The formants F1 and F2 are extracted between the range of 50 and 3000. During this process of data extraction and analysis, the following R packages are used: `emuR` (Winkelmann et al. 2017), `PraatR` (Albin 2014), and `tidyverse` (Wickham 2017). The code and the data used for the analysis are available in the Supplementary Materials.

For the present study, we limited our investigation to 10 languages: 5 Romance languages (Spanish, Catalan, French, Italian and Romanian), 2 Germanic languages (German and Afrikaans), 2 Slavic languages (Polish and Russian) and 1 isolate (Basque). Among them, only two (Spanish and French) can be considered as well-resourced in terms of phonetic data (Mariani et al. 2012; Melero et al. 2012), while the others are mid- to low-resourced, in particular compared to English (Branco et al. 2012; Burchardt et al. 2012; Calzolari et al. 2012; Hernáez et al. 2012; Miłkowski 2012; Moreno et al. 2012; Trandabăț et al. 2012). These languages were selected because (i) they have at least 1000 recordings in Lingua Libre, (ii) they do not use tones as a distinctive feature, (iii) they do not use vowel harmony (i.e., levelling in the quality of all the vowels in a word), (iv) they are manageable in MAUS, and (v) they have the three vowels /a, i, u/ in their MAUS inventory (for instance, English had enough data in Lingua Libre and a segmentation tool in MAUS, but the vowel /a/ was not taken into account by the MAUS inventory for English, which could only align /æ/ and /ɑ/ for this language). During the analysis, we did not include metadata such as gender and geographic location due to two reasons. First, these elements are not required by the recording system of Lingua Libre. Therefore, it is not systematically provided by speakers and a large portion of the speakers are not annotated with the information of gender and/or geographical location. Second, for speakers with this metadata, the information also requires manual cleaning. As an example, speakers provided geographical location at different levels, including towns, cities, and countries. Some speakers also provided non-existing location names. Due to these reasons, we did not include metadata in the current analysis. However, we plan on including them in future analyses after a process of data wrangling.

In the following, we provide a short typological description for each language and the number of distinctive vocalic qualities, or vocalic timbres. Languages are listed from the largest number of vocalic timbres to the smallest. Note that we decided to range languages according to their vocalic timbres, not their actual vowel inventory, since vowel inventories can vary from author to author (see e.g., Moran and McCloy (2019)). To do so, we used the inventory of each language as displayed in WebMAUS and manually counted the number of different vocalic timbres. During this process, long and short vowels were merged and nasal and rhoticized vowels as well as diphthongs and triphthongs were ruled out.

- German (deu): German is a Germanic language mostly spoken in Germany, Austria and Switzerland. It displays 14 vocalic timbres: /i, y, ɪ, ʏ, e, ø, ɛ, œ, a, ə, ɔ, o, ʊ, u/.
- Afrikaans (afr): Afrikaans is a Germanic language mostly spoken in South Africa. It displays 18 vowels: 12 oral /i, y, e, ø, ɛ, œ, a, ɑ, ə, ɔ, o, u/ and 6 nasal /ĩ, ũ, ɛ̃, ɑ̃, ɔ̃, ə̃/, i.e., 12 vocalic timbres.
- French (fre): French is a Romance language mostly spoken in Europe, Northern America and Africa. Standard French displays 16 vowels: 12 oral /i, y, e, ø, ɛ, œ, a, ɑ, ə, ɔ, o, u/ and 4 nasal /ɛ̃, ɑ̃, ɔ̃, œ̃/, i.e., 12 vocalic timbres.
- Catalan (cat): Catalan is a Romance language mostly spoken in Northern Spain. Central Catalan displays 8 vocalic timbres: /i, e, ɛ, a, ə, ɔ, o, u/.
- Italian (ita): Italian is a Romance language mostly spoken in Italy. It displays 7 vocalic timbres: /i, e, ɛ, a, ɔ, o, u/.
- Romanian (ron): Romanian is a Romance language mostly spoken in Romania. It displays 7 vocalic timbres: /i, ɨ, e, ə, a, o, u/.
- Polish (pol): Polish is a Slavic language mostly spoken in Poland. It displays 6 oral vocalic timbres: /i, ɪ, ɛ, a, ɔ, u/ and 2 nasals /ɛ̃, ɔ̃/.
- Russian (rus): Russian is a Slavic language mostly spoken in Russia. It displays 6 vocalic timbres: /i, ɪ, e, a, o, u/.
- Spanish (spa): Spanish is a Romance language mostly spoken in Europe and Latin America. It displays 5 vocalic timbres: /i, e, a, o, u/.
- Basque (eus): Basque is an isolate mostly spoken in Northern Spain and South-Western France. It displays 5 vocalic timbres: /i, e, a, o, u/.

Our vowel inventories therefore range from 5 to 14, with languages across 4 typological groups, although all are Indo-European. Table 1 shows the numbers of [a], [i] and [u]-tokens for the sample of 1000 recordings from each language. The languages are ranked from the largest to the smallest number of oral vowels. The columns [a], [i], and [u] list the raw counts of each vowel across the 1000 recordings extracted for each language. As an example, within 1000 recordings of Basque, the vowel [a] is found 1376 times. The number of speakers is also included. We acknowledge that the number of speakers should ideally be the same across languages, but since the data of Lingua Libre is not controlled as in a formal experiment, we could only aim at having a similar number of speakers per language.

**Table 1.** Count of the vowels [a], [i], and [u] in a sample of 1000 recordings for each language in the data set. The vowel inventory refers to the number of timbres in each language.

| Language | iso | Vowel Inventory | [a] | [i] | [u] | Speaker Count |
|---|---|---|---|---|---|---|
| German | deu | 14 | 551 | 225 | 70 | 9 |
| Afrikaans | afr | 12 | 228 | 249 | 87 | 3 |
| French | fra | 12 | 408 | 285 | 56 | 13 |
| Catalan | cat | 8 | 398 | 478 | 325 | 4 |
| Italian | ita | 7 | 969 | 538 | 104 | 4 |
| Romanian | ron | 7 | 794 | 524 | 377 | 4 |
| Polish | pol | 6 | 936 | 347 | 162 | 8 |
| Russian | rus | 6 | 951 | 560 | 199 | 10 |
| Spanish | spa | 5 | 1087 | 413 | 139 | 13 |
| Basque | eus | 5 | 1376 | 542 | 253 | 6 |

Unsurprisingly, languages with fewer vowel timbres have generally more of each vowel of interest than languages with more vowel timbres. An interesting point is that languages all display more [a]-tokens than [i]-tokens, except for Afrikaans and Catalan, and more [i]-tokens than [u]-tokens. The number of [u]-tokens is rather small for all languages, with only 70 tokens in German (compared to 551 [a] and 225 [i]), and up to only 377 in Romanian (compared to 794 [a] and 524 [i]). To the best of our knowledge, there are no estimations available about the amount of vowels in these vocalic systems, this is thus novel information for linguists, yet in line with the observation that, in vocalic systems, when there is an asymmetry, the number of front vowel types is generally greater than the number of *back* vowel-types (Schwartz et al. 1997). It seems that this observation expands to vowel-tokens.

A sample of the extracted data is shown in Table 2. For each individual occurrence of the vowels [a], [i], and [u], the following information is annotated. First, each identified vowel is assigned a unique identifier so that it is easily extractable from the data. Second, the language of the recording is annotated with its iso-code. For example, French is encoded as 'fra' and German is encoded as 'deu'. Third, the F1 and F2 at the middle point of the occurrence of the vowel are extracted. Then, additional metadata about the speaker ID and the recorded word are provided.

**Table 2.** An example of the data extracted and compiled from Lingua Libre. The rows represent occurrences of the vowels [a], [i], and [u] found in languages of the data set.

| Vowel | ID | iso | F1 | F2 | Speaker | Item |
|---|---|---|---|---|---|---|
| i | 5460 | fra | 430 | 2588 | 0x010C | rime |
| i | 6547 | fra | 315 | 2394 | WikiLucas00 | décider |
| u | 6546 | fra | 356 | 1995 | WikiLucas00 | debout |
| u | 6648 | fra | 323 | 2024 | WikiLucas00 | pelouses |
| a | 5536 | fra | 641 | 2575 | DenisdeShawi | accourent |

Using the extracted formants, the distribution of the vowel space for each language in the data set can then be visualized and analyzed. Taking Italian as an example, we can visualize the distribution of the vowels [a], [i], and [u] in terms of formants in Figure 1. Based on the visualisation, we acknowledge that there are obviously a few outliers in the extracted occurrences. For example, some occurrences of [u] have very different formants from the other occurrences and appear to resemble [ɨ] or [y]. This is likely due to the segmentation accuracy, as some parts of the surrounding context might have been included by the automatic segmentation. These outliers are kept in the raw data. However, they are identified and removed during the analysis. Further details are explained in Section 3.

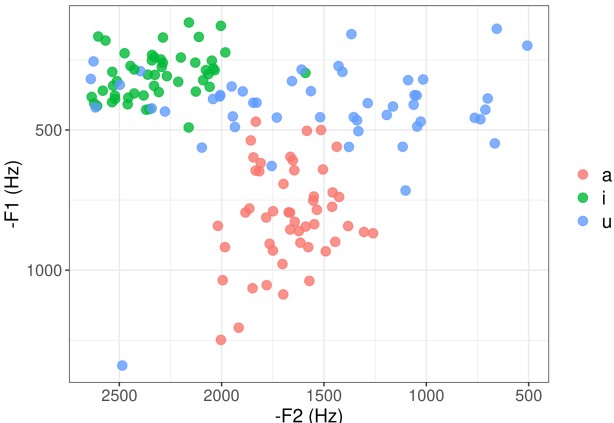

**Figure 1.** The distribution of formants for the vowels [a], [i], and [u] for Italian in the recordings of the data set. Each point represents an occurrence of a vowel in the recordings. The formant values indicate the formants observed at the central point for each occurrence of each vowel.

As a summary, the recordings from Lingua Libre were downloaded. Then, they were automatically segmented using MAUS. This method allowed us to have 50 recordings for each vowel ([a], [i], and [u]) across the 10 languages of the data set. In total, we thus have $50 * 3 * 10 = 1500$ tokens available for analysis.

## 3. Results

First, we visually compare the distribution of formants across each vowel and each language. To facilitate visualization, languages are colored in 3 groups according to their number of vocalic timbres. Languages with more than 10 vocalic timbres are (arbitrarily) affiliated to the group 'large'. Languages with less than 10 vocalic timbres but more than 6 are (arbitrarily) affiliated to the group 'medium'. Languages with 6 or less vocalic timbres are (arbitrarily) affiliated to the group 'small'.

As shown in Figure 2, F1 does not seem to display more variation in languages with smaller vowel inventories in any vowel. F2 values however seem a little more dispersed for [a] in large and small inventories but less in medium inventories for any of the vowels. F2 values for [i] seem to display a little more variation in Romanian, a medium inventory language, and Russian, a small inventory language. As for [u], F2 values indeed seem more dispersed for languages with small vowel inventories. As such, these results do not provide evidence in favor of the ISH.

Considering the variation of F1 and F2 separately does not display directly the variation of the acoustic space across vowels and languages. Therefore, we conduct a 2D kernel density estimation (Venables et al. 2002, R function kde2d) and show the results with contours. We use this method since it allows us to visualize the area covered by all the data points and to simultaneously take into account the density of the data points and remove outliers. As a way to remove the potential outliers mentioned in Figure 1, the number of contours are set to 4, so that outliers likely induced by segmentation errors are not taken into account when measuring the variation of vowels in different languages. To make the

visualization less crowded, we look at the three vowels [a], [i], and [u] separately. First, we visualize the distribution of the occurrences of the vowel [a] as density plots in Figure 3.

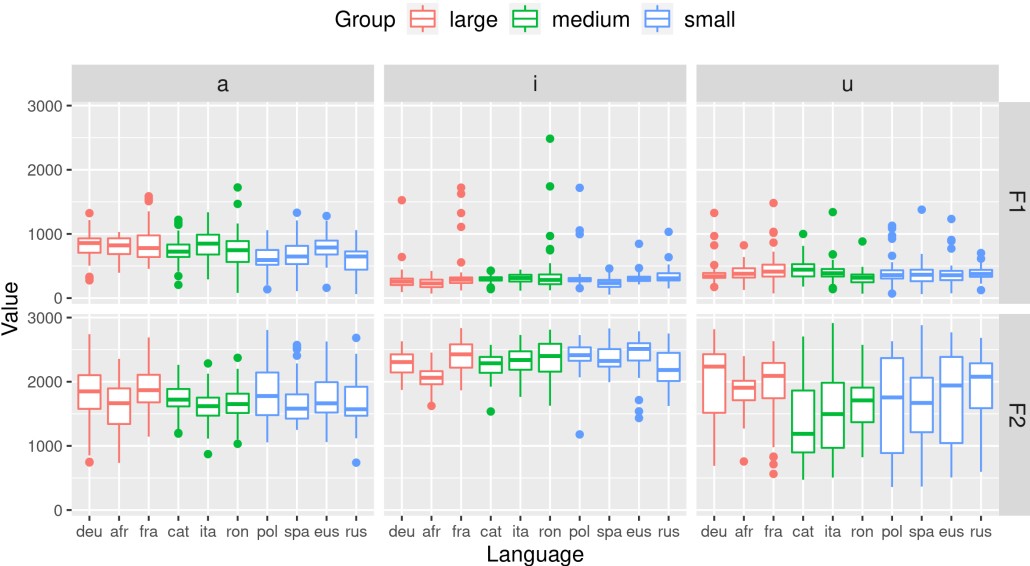

**Figure 2.** The distribution of formants for [a], [i] and [u] across the 10 languages of the data set. The languages on the x-axis are ranked according to their number of timbres from high to low. The colors indicate the group to which the language is affiliated with in terms of number of timbres: red = large, green = medium, blue = small.

In Figure 3, we can see that the dispersion of [a] realizations seems reduced for Catalan and Italian, i.e., some languages of the data set having a medium vowel inventory. This result is surprising with regards to the ISH, since we would rather expect this behavior in languages with a large inventory. However, as expected, languages with a smaller vowel inventory seem to display more dispersion than the other languages, especially Polish.

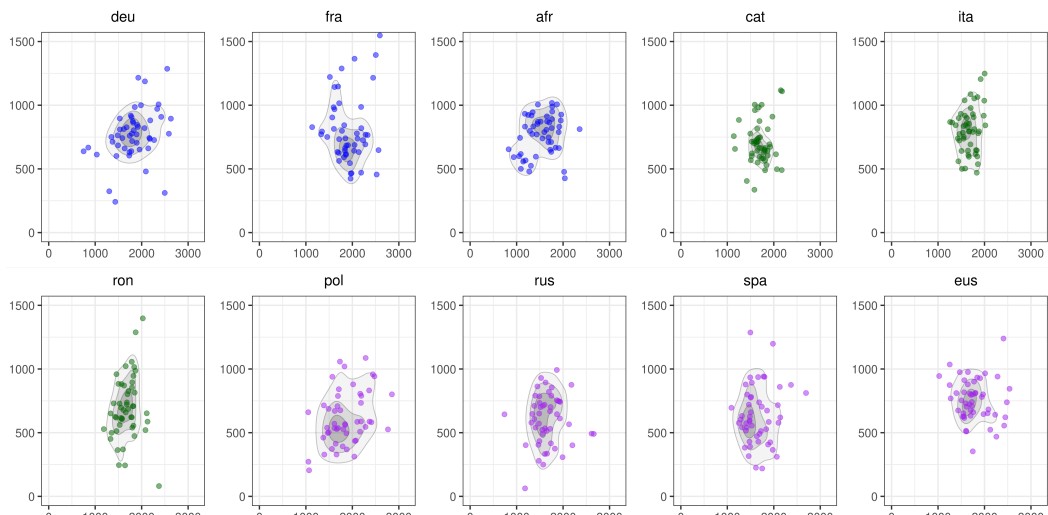

**Figure 3.** The distribution of formants for the vowel [a] across the 10 languages of the data set. The x-axis represents F2 and the y-axis indicates F1. The labels on the top of each sub-figure refer to the iso code of the 10 languages in the data set. The colors indicate the group to which the language is affiliated with in terms of number of timbres: blue = large, green = medium, purple = small.

In Figure 4, we consider the vowel [i]. The realization seems the least dispersed in Polish but the most in Romanian, the former having a small inventory and the latter a medium inventory.

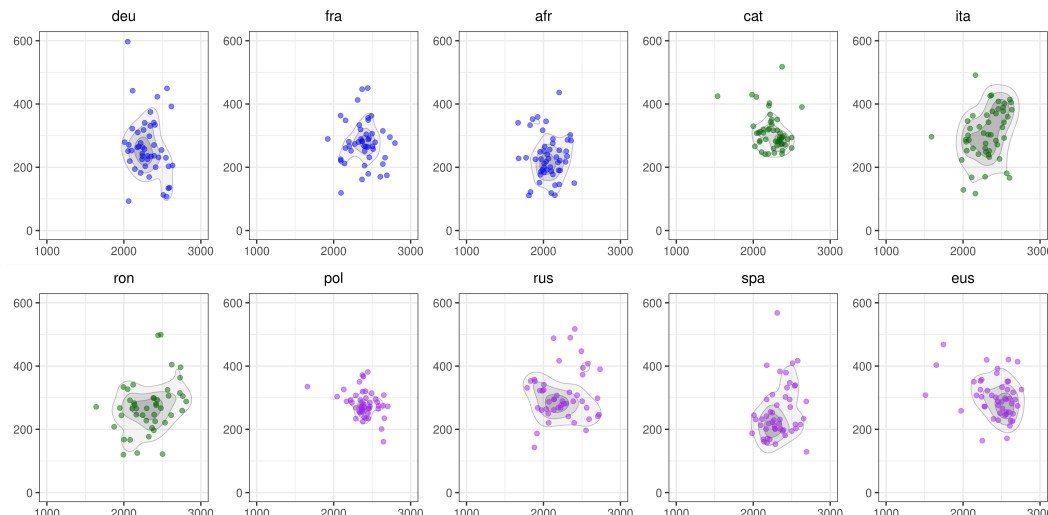

**Figure 4.** The distribution of formants for the vowel [i] across the 10 languages of the data set. The x-axis represents F2 and the y-axis indicates F1. The labels on the top of each sub-figure refer to the iso code of the 10 languages in the data set. The colors indicate the group to which the language is affiliated with in terms of number of timbres: blue = large, green = medium, purple = small.

In Figure 5, we visualize the distribution of formants for the vowel [u]. Patterns of realizations differ from one language to the other, but in general, no tendency emerges. French and German, which have a large vowel inventory, also display a lot of variation, which is contrary to the ISH. So do Catalan and Italian, which have 8 and 7 vowels in their respective inventories, but not Romanian, which has also 7. Finally, we find the expected dispersion in almost all languages with a small inventory.

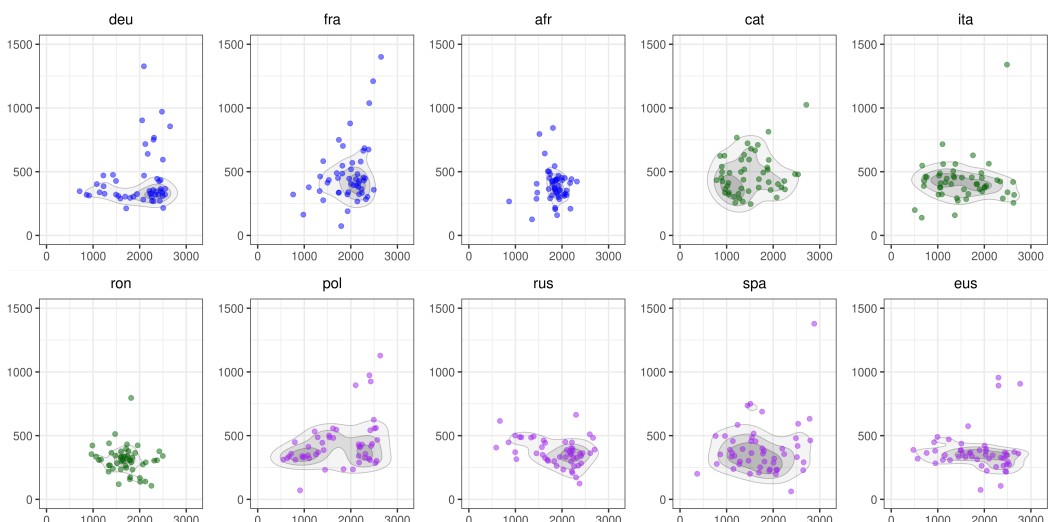

**Figure 5.** The distribution of formants for the vowel [u] across the 10 languages of the data set. The x-axis represents F2 and the y-axis indicates F1. The labels on the top of each sub-figure refer to the iso code of the 10 languages in the data set. The colors indicate the group to which the language is affiliated with in terms of number of timbres: blue = large, green = medium, purple = small.

The visualization is inconclusive regarding potential effects of the ISH. Additional testing is thus required to assess the statistical significance of these observations.

To do so, we extract the areas of each vowel in each language and run linear mixed models (Kuznetsova et al. 2017) to test the ISH. These models are commonly used in linguistic studies when controlling for random effects is necessary (Bentz and Winter 2013; Ladd et al. 2015; Sinnemäki and Di Garbo 2018). More precisely, linear mixed models predict

the value of a dependent variable based on several predictor variables, while also taking into account the effects of random groupings. In the current paper, we run three models with three different dependent variables. Each of the three models uses the number of timbres, the inventory size groups (large, medium or small), the vowel (i.e., [a], [i], and [u]), and the presence/absence of schwa in the phonemic inventory[2], to predict the dependent variable given the random structures found across languages and replications. The first model takes the areas covered by each vowel in each language as dependent variable. The second model considers the standard deviation of F1 as the dependent variable. The third model considers the standard deviation of F2 as the dependent variable. We expect that these three models will show different perspectives of the variation in vowel production. The area considers the acoustic space as a whole, while F1 and F2 represent the two axes of the acoustic space. As an overview, the ISH predicts a negative association between number of timbres and the dependent variables since it predicts that the area covered by the variation of a vowel is smaller if the language has a larger number of timbres. To enhance the robustness of the models, we also conduct 10 samplings of vowel tokens. That is to say, we consider the possibility that the sample of 50 tokens per vowel and language could be influenced by randomness and not be stable across several samplings. To be more precise, the areas covered by the formants could vary a lot across different samplings of vowels. Therefore, we conduct 10 samplings and fed the output of all 10 samplings to the three mixed models. The results of the three models are shown in Table 3.

**Table 3.** The output of linear mixed models based on the output of 10 vowel samplings with 50 tokens for each vowel in each language. The abbreviations are read as follows: nb_of_timbres = number of timbres, sd = standard deviation. The significance codes are interpreted as follows: * indicates a *p* value < 0.05, *** indicates a *p* value < 0.001.

| Dependent Variable | Predictor | Estimate | df | t Value | *p* Value |
|---|---|---|---|---|---|
| Area | nb_of_timbres | 0.002 | 5 | 0.038 | 0.972 |
| Area | vowel_i | −0.064 | 288 | −1.437 | 0.152 |
| Area | vowel_u | −0.077 | 288 | −1.731 | 0.085 |
| Area | Group_medium | 0.040 | 5 | 0.154 | 0.883 |
| Area | Group_small | 0.084 | 5 | 0.243 | 0.818 |
| Area | with_schwa | −0.007 | 5 | −0.057 | 0.957 |
| sd F1 | nb_of_timbres | −20.387 | 5 | −0.732 | 0.497 |
| sd F1 | vowel_i | −61.511 | 288 | −7.385 | 0.000 *** |
| sd F1 | vowel_u | −50.488 | 288 | −6.062 | 0.000 *** |
| sd F1 | Group_medium | −99.208 | 5 | −0.648 | 0.545 |
| sd F1 | Group_small | −112.119 | 5 | −0.552 | 0.605 |
| sd F1 | with_schwa | 57.125 | 5 | 0.804 | 0.458 |
| sd F2 | nb_of_timbres | 40.694 | 5 | 3.197 | 0.024 * |
| sd F2 | vowel_i | −114.216 | 288 | −11.013 | 0.000 *** |
| sd F2 | vowel_u | 225.373 | 288 | 21.731 | 0.000 *** |
| sd F2 | Group_medium | 191.031 | 5 | 2.732 | 0.041 * |
| sd F2 | Group_small | 329.166 | 5 | 3.548 | 0.016 * |
| sd F2 | with_schwa | −51.890 | 5 | −1.599 | 0.171 |

In general, we observe that the vowel space of the vowels [a], [i], and [u] is not significantly associated with the number of timbres, neither with the inventory group, nor with the presence/absence of schwa. In terms of formants, there are diverging results between F1 and F2. For F1, no significant effect is found for the number of timbres nor the presence/absence of schwa. The vowel [a] tends to have a larger standard deviation for F1 than the other vowels. In terms of F2, the vowel [u] tends to have a larger standard deviation, while the vowel [i] tends to have a smaller standard deviation. The presence/absence of schwa does not have a significant effect on F2 either. However, both number of timbres and inventory groups have significant effects on F2, but number of timbres has a positive correlation with F2 standard deviation (i.e., the more timbres, the more variation in F2),

which invalidates the ISH, while groups by inventory size have a negative correlation (i.e., the larger the group, the less variation in F2), which partially validates the ISH. This is probably due to the fact that the model controls for number of timbres within each group. While the standard deviation of F2 is negatively correlated with number of timbres when mixing all data points, it is positively correlated with number of timbres within languages from the same group. Hence, when only considering the number of timbres in each language, without artificially grouping languages with regards to their inventory size, our results do not support the ISH, even on F2 specifically.

Finally, we also run tests that do not consider random structures in the data. These tests include: comparison of quantiles and comparison of variation coefficients. The results of these tests generally match the visualization in Figure 2 and the output of our mixed models. They suggest that there is more variation in F2 than in F1, and that the variation across groups has a larger magnitude in the vowel [u]. Additional details from the output of all the models and tests are available in the Supplementary Materials.

## 4. Discussion

In this paper, we have provided the first phonetic study using the crowd-sourced data from Lingua Libre to test the Inventory Size Hypothesis on a substantial amount of languages. We tested 10 languages, among which 8 can be said to be less-resourced. The data was automatically segmented using WebMAUS and analyzed using R. Two types of visualization (boxplots and scatter plots), a linear mixed model and various complementary statistical tests all fail to confirm that a larger vowel inventory correlates with smaller dispersion in production. This failure supports the results of the previous recent studies that handled a substantial amount of languages showing that the ISH is not accurate to account for phonetic variation, at least not in this form (Hutin and Allassonnière-Tang 2022b).

However, several limitations should be taken into account. First, the study focuses on only 10 languages (out of approximately 7000 estimated worldwide) and all are Indo-European. Second, it is possible that the interaction, if any, between inventory size and dispersion of realizations is more complex than what meets the eyes. For instance, /a/ behaves differently than /i, u/ regarding F1, and /u/ than /a, i/ regarding F2. This may be why the mixed effects models fail to detect an interaction. It is also possible that the model is not fed with enough data, since we worked with only 1500 tokens. Finally, it is possible that the contributors may be inclined to hyper-articulate when recording prompted isolated words for patrimonial purposes, and thus display less variation than in natural connected speech.

Our study thus indicates that the ISH is inaccurate, which confirms the literature, but it may also need further investigations. Now that this proof of concept has proven useful, we would like to extend our methodology to more languages from more diverse language families. This will be possible in the near future since Lingua Libre is growing rapidly (in March 2022, the library contains almost 686,000 recordings, i.e., 9000 more than in February of that same year). We would also like to extend our study to more connected speech, which is now possible since Lingua Libre was recently updated to offer the possibility to adjust the time needed for the device to detect the pause, thus allowing to read longer texts, even with prosodic pauses between phrases or even paragraphs.

We would like to finish by providing some conclusions on the main topic of this paper, i.e., the use of crowd-sourced data from Lingua Libre to investigate scientific research questions. As summed up above, the general methodology is promising, since the results are in line with previous literature using professional corpora. The comparability of Lingua Libre and professional corpora has also been demonstrated in another study on the particular case of Polish (Hutin and Allassonnière-Tang 2022a).

On a positive note, our study shows that Lingua Libre has an impressive potential. First, it already provides great amounts of data in many languages, and as an open-access, crowd-sourced tool, it has the advantage of offering an ever-growing data set, which may help bridge the gap between well- and less-resourced languages. Second, the quality of the

recordings is sufficient for extracting acoustic measures, which is ideal for phonetic research. Finally, the documentation to access and exploit it is easily accessible and usable, which makes its exploitation feasible by any researcher. However, we also came across several difficulties, in particular regarding the metadata. First, the names of the files were not always following the same template, some following the regular "language-speaker-item" template, such as "ita-Yiyi-Brissago-Valtravaglia", others following a different template, such as "Pamputt-fra-lapin-garou". This problem made the automatic processing of files complicated. Second, linking the data to the speakers' metadata proved difficult, since the process has not yet been automatized in Lingua Libre, and we had to process it manually. Moreover, the metadata were not always accurately completed by the participants. Our suggestions for improvement would thus be to improve the uniformity of the file names and the processing of the metadata. This implies not only facilitating the linking between files and metadata, but also raising the contributors' awareness on the need for scientists to have access to accurate, clean information. A verification tool *à la* Common Voice could also improve the quality of both recordings and metadata. Another difficulty arose not from Lingua Libre *per se* but from the limits of our methodology for segmentation. We were indeed limited by MAUS to languages handled by the software. Future research would also benefit from investigating more alignment tools to allow the investigation of more diversified languages, from more language families and with more various phonemic inventories.

## 5. Conclusions

All in all, since Lingua Libre proved useful to investigate a specific research topic, we hope to use it again for future research. On this same research topic, we would like to investigate more linguistic variables (such as the duration and F3, F4, F5 of the vowels, their position in the syllable and word, whether they are bearing lexical accent or not...), as well as sociolinguistic variables (such as gender, geographical location, etc.). But most of all, obviously, we hope that the data set will continue to grow rapidly so that we can include more and more languages to our study.

**Supplementary Materials:** The code used in this study can be downloaded at https://github.com/marctang/Operation_Lili, accessed on 30 August 2022.

**Author Contributions:** Conceptualization, M.H. and M.A.-T.; methodology, M.H. and M.A.-T.; validation, M.H. and M.A.-T.; formal analysis, M.H.; investigation, M.H. and M.A.-T.; data curation, M.H. and M.A.-T.; writing—original draft preparation, M.H. and M.A.-T.; writing—review and editing, M.H. and M.A.-T. All authors have read and agreed to the published version of the manuscript.

**Funding:** M.H. is thankful for the support of the project OTELO (OnTologies pour l'Enrichissement de l'analyse Linguistique de l'Oral (PI Ioana Vasilescu and Fabian Suchanek), supported by the Excellency Award of Institut DATAIA and the MSH Paris-Saclay. M.A.-T. is thankful for the support of the French National Research Agency, grant EVOGRAM: The role of linguistic and non-linguistic factors in the evolution of nominal classification systems, ANR-20-CE27-0021 (PI M.A.-T.).

**Institutional Review Board Statement:** Not applicable.

**Informed Consent Statement:** Not applicable.

**Data Availability Statement:** The data used in this study is publicly available at https://lingualibre.org, accessed on 30 August 2022. The Supplementary Materials of this study can be found at the following link: https://github.com/marctang/Operation_Lili, accessed on 30 August 2022.

**Acknowledgments:** Our heartfelt gratitude goes to the three anonymous reviewers who have been extremely insightful and helpful in improving the paper. We would also like to thank the Wikimedia community, and in particular Lucas Lévêque, who provided precious insight on the Lingua Libre platform, as well as Lucas Ondel (CNRS) for suggestions on the formant analysis.

**Conflicts of Interest:** The authors declare no conflict of interest.

## Abbreviations

The following abbreviations are used in this manuscript:

| | |
|---|---|
| VDT | Vowel Dispersion Theory |
| ISH | Inventory Size Hypothesis |
| MAUS | Munich Automatic Segmentation System |
| SAMPA | Speech Assessment Methods Phonetic Alphabet |

## Notes

[1]  The data of Lingua Libre can be downloaded directly at https://lingualibre.org/datasets/, accessed on 30 August 2022.

[2]  Since schwa is by definition the central vowel in inventories, it is expected, if the ISH is valid, to repel other vowels to the periphery of the acoustic space.

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
