# Peer review of "Operation LiLi: Using Crowd-Sourced Data and Automatic Alignment to Investigate the Phonetics and Phonology of Less-Resourced Languages"

_languages, doi:10.3390/languages7030234_

Round 1
Reviewer 1 Report
This a well-written paper that discusses the use of crowd-sourced data to investigate vowel variation, vowel categories and vowel inventory size in 10 languages to see if their data provide support for the Inventory Size Hypothesis. They have carried out automatic alignment of the vowels. The results show no support of the Inventory Size Hypothesis in line with the findings of several other studies.
Specific comments
Abstract
Rewrite that two last sentence like this?
“Information on the formants of the vowel segments is then extracted to measure how vowels expand in the acoustic space and whether this is correlated with the number of vowel categories in the language. The results provide valuable insight into the question of vowel dispersion and demonstrate the wealth of information that crowd-sourced data has to offer.”
Page 2
Lines 40+41: “expand within a larger acoustic space”, “expand within a smaller acoustic space”
Page 3
Line 106: -> the recordings are selected
Line 108: -> some languages have many more datapoints
Line 109: -> The threshold 50 was chosen
Page 4
Line 140: -> In the following, we provide a short typological description for each language and the …
Page 5
Line 179: -> languages with fewer vowel timbers
Page 6
The article has both UK and US spellings, e.g. ‘colored’ (line 211) and ‘behaviour’ (line 233). Go for one standard spelling. This must be checked throughout the article.
Page 11
Line 300: -> This failure supports
Line 309: > languages from more diverse language families. This will be possible in the near future since Lingua Libre is growing rapidly
Line 313: -> We would like to finish by providing
Line 320: -> and less-resourced languages.
Line 330: -> not always accurately completed by the participants.
Line 340: -> proved useful to investigate
Line 343: I would delete the sentence “Several submitted and even accepted concomitant works go in this direction”
Reviewer 2 Report
This paper provides an excellent overview of how Lingua Libre, a crowd-sourced corpus of cross-linguistic spoken audio, can be used for advancing questions related to phonetic typology. In particular, the paper investigates a nicely defined hypotheses, the Inventory Size Hypothesis, which posits that as the size of a vowel inventory increases, the corresponding variance of those vowels should decrease. The paper examines data from 10 languages which might be considered less studied or lower resourced than English. One limitation of the current study is that its end goal ultimately does seem to be about phonetic typology, but the current sample includes data from European (or Indo-European) languages only. Nonetheless, I think the paper provides an excellent foundation for further work in this area and with this corpus. With some minor revisions, I think this is a nice piece of work.
Questions and comments:
It will be necessary to rephrase or remove the following sentence as it is noticeably similar to a tweet from Eleanor Chodroff: “A major challenge in data-driven research on typology is having sufficient amounts of data in a sufficient number of languages to draw meaningful conclusions”. (Original sentence from Eleanor Chodroff: “A major hurdle in data-driven research on typology is having sufficient data in a large number of languages to draw meaningful conclusions.”)
Ultimately ISH seems to be a typological claim, but only a limited number of Indo-European languages are tested here. The study is really excellent and I'm excited to see the hypothesis tested on a larger number of languages in a future work, but this current limitation should be discussed.
The Universal Dependencies Corpus is not typically considered a phonetic corpus, so I'm somewhat confused by its mention in the introduction. Could this be clarified or revised to embed it in a clearer manner?
How is less-resourced defined? It would be good to add this definition. If the comparison is English and Mandarin/Chinese, then wouldn't every language be less-resourced? I then don't think the claim in the abstract that phonetic studies focus only on English or Mandarin is entirely true, regardless of whether "large corpora" there means multilingual large corpora or even monolingual large corpora. Then again, in the introduction, two of the languages are explicitly described as not less-resourced.
In the introduction, another relevant paper and dataset alongside Vox Clamantis is the Vox Communis Corpus, a derivative phonetic corpus from the mentioned Common Voice Corpus. This is from Ann & Chodroff (2022), published at LREC.
I'm a bit confused how you were able to get so much data from each language. When I go to LinguaLibre.org, I then went to Datasets, and it looks like the Afrikaans zip file has only 3 speakers in it, not 8. Is there another location where the data was downloaded from? (On a second visit, I see there's a zip file with the word "full" in it: if that was the downloaded data set, it might be good to mention this for ease of replicability, and especially if one of the main goals of the paper is to promote the corpus.)
Relatedly, on p. 3, it says that 10 speakers were used from each language, but Table 1 has speaker counts ranging from 5 to 14. Perhaps the page 3 sentence should include the word "approximately" (listing such a precise number could be a bit confusing).
A few methodological suggestions and questions:
- can you describe what is contained in each audio file a little more clearly? Is it a single word or a spoken sentence (I believe it's a single word)?
- It would be good to add the formant extraction settings (e.g., floor and ceiling values). Supplementary materials are mentioned, but I don't think I currently have access to those, so I can't check this.
- How was gender handled? Gender can have huge influences not only in how the formants are extracted, but also in how it should be analysed. How was this handled in extraction, and secondly, how was this handled in the model?
- How was dialect handled (e.g., for Spanish which can vary widely depending on which dialect you're dealing with; same for German and French)?
- Can you add the motivation for kernel density estimation? Also, how does it work, and what was the ultimate criterion for outlier exclusion?
- Was a random intercept for speaker also included in the model? It could be that many of the sampled speakers of language X simply had larger areas than many of the sampled speakers of language Y, and then findings would tell us more about the individual speakers than about the languages. A random intercept for speaker should be included in this model.
This may not be necessary but would be really great -- would it be possible to include a figure of the averaged vowel area between [i a u] in F1 x F2 space for each of the 10 languages?
Typo: Saleski -> Salesky
Typo: timber -> timbre or vowel quality or vowel category
Reviewer 3 Report
Thank you for this interesting article. I have a few general questions that might help to improve the quality of this study when being addressed in your text, followed by some minor remarks:
1. Lingua Libre is a fascinating source, but in the context of your ISH I see a problem: Lingua Libre contains only recording of single read words, and thus the words will be articulated in a 'citation mode' as opposed to real fluent speech as in a conversation (or even a read continuous text). I'm a phonetician, therefore I would expect the vowel space from vowels taken from this kind of recordings to be quite different from vowels taken from real speech (the former being more on the 'hyper' end of the HH continuum, and the latter at the 'hypo' end). You do not address this problem in your text; can you give any arguments why recordings of read single words should behave with regard to the vowel dispersion like vowels from real speech?
2. There is something wrong with your phoneme encoding throughout the text. In line 144 you say that you are using the SAMPA inventory (without giving a reference for SAMPA BTW), but in the then following lists you are using a mix of SAMPA and IPA, e.g. "/i, y, I, Y, e, ø, E, oe, a, @, O, o, U, u/" where ø and oe are IPA and the rest is SAMPA; the same is to be said when you list nasalized vowels: "/˜E, A˜, ˜O, oe˜/" which are shown in my viewer with the tilde above the symbol (as in IPA) while in SAMPA the tilde is after the character. Please carefully examine all these cases and replace then by proper SAMPA symbols.
3. Vowel dispersion in the F1/F2 space is dependent on the syllable position of the vowel, e.g. word-final vowels tend to be more centralized than word-initial vowels etc.
I don't find any information about syllable position in the text, but I gather from the few examples in Table 2 that you did not control for syllable position. Can you justify this? When using only 50 vowel tokens this surely is a factor that cannot be neglected.
4. Lines 252-255
"Each of the three models uses the number of timbers, the inventory size groups (large, medium or small), the vowel (i.e., [a], [i], and [u]), ... to predict the dependent variable given the random structures found across languages and vowels."
There are several things here I don't understand:
If the vowel class is a random factor, how can you use it again as a predicting factor?
And why would you apply the vowel class as a predicting factor at all? It is clear that this factor will be highly significant in the model since the vowel formants are quite different per se, and it will make the model more complex without any benefit.
The two factors 'number of timbers' and 'inventory size group' are dependent of each other; you should not use dependent variables as predicting factors in the same model.
To make the discussion easier: could you simply provide the formulas of these models?
5. Lines 268-269 :
"Therefore, we conduct 10 samplings and fed the output of all 10 samplings to the three mixed models."
I'm not a trained statistician but I'm pretty sure that it is not allowed to mix 10 random samplings in one pool of data and then run an analysis across the data in this pool ('repeated measures'!). Or did you use the number of sampling bin (1-10) as a random factor in your model?
Why don't you simply enlarge the sampling size?
Minor remarks:
- in the manuscript lines 36-42 extend beyond the margin (and later on again, e.g l. 126, 224ff etc.; I tried several PDF viewers but the problem remains)
- the Reference section contains a lot of typos/format errors;
I would suggest to replace [27] by the correct following reference (found on the WebMAUS webpage):
Florian Schiel (1999): Automatic Phonetic Transcription of Non-Prompted Speech, Proc. of the ICPhS 1999. San Francisco, August 1999. pp. 607-610.
And if you are looking for a real reference for the EMU-SDMS in [32] you might consider using the following (taken from the EMU-SDMS web page):
Raphael Winkelmann, Jonathan Harrington, and Klaus Jänsch, (2017): EMU-SDMS: Advanced speech database management and analysis in R Computer Speech & Language, Volume 45, 392-410.
- l. 113 : "MAUS" is an abbreviation (used later in the text) and should therefore be in brackets: "Munich AUtomatic Segmentation System (MAUS)"
- l. 122 : "For each recording of each vowel, the F1 and F2 at the middle of the entire sound are extracted."
Could you be a bit more specific here?
What algorithm/tool did you use, one of praat or of emuR? Did you extracted the F1/F2 values as an average over a certain middle proportion (say 20%) of each segment or did you really use the exact midpoint value taken from the formant trajectory (the latter is problematic as it can more often be an out-layer value!)
- l. 133 "they have at least 100 recordings'
Do you mean '1000 recordings' here? Because earlier/later you always postulate 1000 recordings....
- l. 247 "To do so, we extract the areas of each vowel in each language..."
Which area precisely? Please give the formula.
Round 2
Reviewer 3 Report
The paper was improved considerably, thank you.
Just a few final answers to your comments:
- I'd suggest to omit reference [28]. This is a very specialized paper regarding an extension of the MAUS package that is not actually used in WebMAUS.
- the text extending beyond margins is not visible in my PDF viewer any longer.
- you replaced all the inter-mixed SAMPA symbols by the correct IPA symbols; this looks fine now.
Author Response
Dear reviewer,
We have deleted the reference to Schiel (2004).
We hope the revised version attached complies with your last comments.
Kind regards,
The authors
